# Trends of Hydroclimatic Intensity in Colombia

Oscar Mesa *,†, Viviana Urrea † and Andrés Ochoa †

Departamento de Geociencias y Medio Ambiente, Universidad Nacional de Colombia, Carrera 80 # 64-223, 050041 Medellín, Colombia; vurream@unal.edu.co (V.U.); aochoaj@unal.edu.co (A.O.)
* Correspondence: ojmesa@unal.edu.co; Tel.: +57-300-787-4736
† These authors contributed equally to this work.

**Abstract:** Prediction of precipitation changes caused by global climate change is a practical and scientific problem of high complexity. To advance, we look at the record of all available rain gauges in Colombia and at the CHIRPS database to estimate trends in essential variables describing precipitation, including HY-INT, an index of the hydrologic cycle's intensity. Most of the gauges and cells do not show significant trends. Moreover, the signs of the statistically significant trends are opposite between the two datasets. Satisfactory explanation for the discrepancy remains open. Among the CHIRPS database's statistically significant trends, the western regions (Pacific and Andes) tend to a more intense hydrologic cycle, increasing both intensity and mean dry spell length, whereas for the northern and eastern regions (Caribbean, Orinoco, and Amazon), the tendencies are opposite. This dipole in trends suggests different mechanisms: ENSO affects western Colombia more directly, whereas rainfall in the eastern regions depends more on the Atlantic Ocean, Caribbean Sea, and Amazon basin dynamics. Nevertheless, there is countrywide accord among gauges and cells with significant increasing trends for annual precipitation. Overall, these observations constitute essential evidence of the need for developing a more satisfactory theory of climate change effects on tropical precipitation.

**Keywords:** precipitation trends; climate change; colombia

## 1. Introduction

Predicting the effect of climate change on Colombia's hydrology, specifically precipitation, is not a minor issue. To illustrate, only in the electricity sector recent studies for the Colombian Mining and Energy Planning Unit [1] estimate that the impacts of a decrease in precipitation would imply an increase in annual investment of US$ 290 million per year for the period 2013–2015. The explanation for this increase in investments is that hydroelectric generation meets approximately 70% of the country's electricity demand. We will show that this alleged declining trend obtained from models does not correspond with past observations.

Even though our paper is focused on analyzing statistical trends in precipitation records, the following broad theoretical considerations are an essential framework for these observational analyses. First, there are critical theoretical reasons besides the obvious practical importance of understanding and predicting the effect of climate change on Colombia's rainfall. General Circulation Models (GCMs) are the only prediction tools for assessing climate change effects on tropical precipitation. However, they differ substantially across the different models [2]. Additionally, despite some building blocks for the construction of a theory of tropical precipitation change, there is no satisfactory one yet.

Some of the building blocks mentioned above are the decomposition between "thermodynamic" (due to changes in atmospheric moisture), and "dynamic," (due to changes in the intensity and location of atmospheric circulation features) [3]. A further decomposition of the dynamic term is between a "weakening" term (due to the slowing down of atmospheric circulation under global warming), and a "shift" term (due to precipitation

patterns change). The thermodynamic term is due to changes in surface-specific humidity. Over oceans, the change in surface humidity observations indicates a near-constant relative humidity. Together with the warming trend and the consequent enhanced water holding capacity of air due the Clausius–Clapeyron equation, this produces an estimation of an increase of about 7% $K^{-1}$. Over continents, the increases may be smaller depending on the possible reductions of relative humidity.

Another building block is the concept of the "rich-get-richer" response, which predicts an increase in precipitation of about 3% $K^{-1}$ over land regions of climatological atmospheric convergence and a small reduction or no change in regions of divergence [4]. These ideas relate the Inter-Tropical Convergence Zone (ITCZ) location to interhemispheric contrasts in temperature and net radiative fluxes at the top of the atmosphere. The critical point is that warming or cooling one hemisphere relative to the other would require an unreasonable cross-equatorial flow of energy into the colder hemisphere and an ITCZ shift.

However, the ITCZ position is not the only factor; El Niño-Southern Oscillation (ENSO) is another key player of the tropical climate. What are the effects of climate change on ENSO? Are warm events becoming more or less intense? Is there a change in the spatial pattern of the warming of the tropical Pacific? Are the central Pacific events becoming more frequent that the eastern Pacific ones? There is no consensus about these fundamental issues that affect the global climate and hydrology. See Rojo Hernández et al. [5] and references cited there for a recent review. As ENSO is an essential control of Colombian hydrology [6], one expects that possible trends in ENSO frequency or space pattern will imprint on the possible trends in Colombian rainfall.

Moreover, Colombian climate and hydrology are not as simple as being controlled by ENSO alone. The Caribbean Sea, the Atlantic Ocean, and the Amazon basin are also critical macro-climatic factors influencing moisture influxes and, therefore, rainfall. Some studies based on models predict a reduction in Amazon's rain because of land surface process changes associated with $CO_2$ concentration increase [7]. Are there regional patterns in the Colombian rainfall trends that may show that the Pacific trend influences some regions with a common sign? Do other regions' trends exhibit a different trend with more influence from the Atlantic or the Amazon?

In addition to precipitation, global change impacts many more aspects such as temperature, sea level, coastal erosion, paramo ecosystem loss, vector-borne diseases, biodiversity, agriculture, and others.

In addition to the difficult theoretical framework discussed above, adequate observations of the complexity of rainfall fields and tropical climate in the rough topography of the Andes cordillera is also a major difficulty: rainfall records in Colombia are generally scarce because of their quality, missing data, length of the records, and spatial coverage. To address that issue, we use two datasets, rain gauges and the satellite product CHIRPS [8]. Rain gauge data refer to point measurements, while CHIRPS data (a gridded dataset) provide precipitation covering larger geographical areas (one cell is 30.86 $km^2$). CHIRPS blends data from infrared cold cloud duration satellite observations with station data. The data quality issue may reflect possible concordance or not between trends estimated from point rainfall records and the CHIRPS database. Moreover, the irregularity of the rainfall fields constitutes a challenge in the statistical estimation of trends and spatial patterns of trends. Contrasting both databases' results may provide clues that may help future theories of the space-time structure of rainfall fields [9].

This paper focuses on examining the trends in the observational record of precipitation over Colombia as a necessary first step to answer these essential practical questions. Additionally, the evidence of trends in the historical record of precipitation may contribute to understanding the impact of climate change on precipitation. We first present a brief review of previous work, followed by a description of the data and the methods. The remainder of the paper presents and discusses the results.

### 1.1. Previous Work

This short review has two parts. First, we present the main results of previous studies about climate change impact on Colombian rainfall trends. Then we briefly show the general context of how global warming impacts precipitation.

#### 1.1.1. Colombian Rainfall Trends

Various works describe the climatology of the precipitation in Colombia [10–15]. The central control is the passage, twice a year, of the ITCZ that marks the rainy seasons of April-May and September-November in the Andes, and the seasons with the lowest rainfall in December-February and June-August. The spatial distribution of precipitation is controled by the sources of humidity in the Caribbean, the Pacific, and the Amazon, by the topography and prevailing winds. Three low-level jets play a significant role, namely the Caribbean, Chocó and the jet along the eastern Andes South America. The inter-annual variability is controlled mainly by ENSO's phenomenon in the tropical Pacific [6,16].

Several studies have found evidence of climate change in Colombia using various statistical techniques with different record lengths [13,17–25]. In summary, these studies identify increasing mean and minimum temperature records in a significant number of stations. Moreover, they find mixed trends in precipitation, with a similar percentage of stations for each trend sign and 20% with no statistically significant trend for the set of considered series of up to 40 years of records. For precipitation series with longer records, the majority (63%) shows an increasing trend and only a 16% decreasing trend. There is no clear geographical pattern in the areas with a particular trend, except in the Pacific plain, which has the highest definite upward trend. The explanation for this Pacific trend may come by an increasing trend of the influence of moisture in the Pacific and the Chocó Jet. These conclusions are in agreement with the Colombian Meteorological Service (IDEAM) report, [26], who analyzed 310 rainfall stations with monthly records in the period 1970–2010 using the standardized method RCLIMDEX [27]. They found 71% stations with increasing trend, 22% decreasing trend, and 7% without a trend.

The observed trends may be due to other causes besides increasing greenhouse gas global warming: deforestation and urbanization, among others, not to mention observational issues. Concerning the impact of deforestation, Salazar [28] estimates through a numerical experiment that a possible drastic future change in forest coverage in the Amazon area would bring about a reduction in precipitation in Colombia of about 300 mm/year.

The warming of the Colombian Andes has led to the complete extinction of eight tropical glaciers, and the six remaining snow-caps are losing ice at accelerated rates [29]. The paramos, unique and strategic ecosystems to supply water to several cities, including Bogotá and Medellín, are also in danger by warming and other anthropogenic activities [30].

Mesa et al. [13] and Carmona and Poveda [24] report that a large proportion of the river flow series in the Magdalena-Cauca basins have decreasing trends, whereas 0% to 34% of the analyzed streamflow series show an increase. The positive regional trend for the Atrato and San Juan flows coincide with areas of significant increasing trends in precipitation. In addition to precipitation, trends in river flows may come from evapotranspiration changes.

Hurtado and Mesa [25] developed a reanalysis of the precipitation field in Colombia, comprising 384 fields of monthly precipitation in the period 1975–2006 at a spatial resolution of 5 min of arc. The reanalysis used records of 2270 rain gauges and various satellite-derived products for the most recent period. Then, using Empirical Orthogonal Functions, Principal Component Analysis, and statistical tests, they looked for changes or trends. According to their results, the Mann–Whitney mean change test and the simple $t$ trend test indicate increasing precipitation trends mainly in the Pacific, Orinoco, and Amazon regions. In most of the Andean region, there are no changes or trends.

Ruiz [31] and Pabón [32] analyze the results of the low-resolution GCM's to conclude that "annual precipitation would be reduced in some regions and would increase in others". Most IPCC models predict an increase of around 10% for precipitation over Colombia, except the northernmost zone. The general trends of the individual models or scenarios

agree in sign, although the magnitudes vary. Using downscaling of GCM's, IDEAM [33] predicted decreasing trends that would imply a reduction of precipitation of 20% at the end of the century for many parts of the country. Later [34], this prediction changed to increasing trends throughout most of the country except for the Caribbean region and the southernmost part of the Amazon region, where the prediction remained for a decreasing trend.

Concerning higher time resolution extreme precipitation, Urán [35] carried out an analysis of the scaling between precipitation and temperature limited by the Clausius-Clapeyron equation using 86 rain gauges and nine temperature stations over the Antioquia region of Colombia, with 15 min resolution. He also used rain derived from Tropical Rainfall Measure Mission (TRMM) data with rainfall intensities every 3 h. He found that for temporal scales larger than 12 h, the trends are not significant. However, for the finer temporal scales, trends become significant for extreme deciles of the distribution. He reports a close scaling due to the Clausius-Clapeyron relation limiting the intensification of precipitation following the ideas of O'Gorman and Schneider [36].

### 1.1.2. Impact of Global Warming on Precipitation

In response to global warming, the hydrological cycle also changes. A warmer atmosphere means more radiative cooling of the troposphere, which is a growing function of temperature. The highest infrared radiation emission corresponds to the balance required to compensate for the larger radiation absorbed. Changes in precipitation may occur depending on the extent to which water vapor changes in cloudiness or the absorption of radiation offsets the necessary radiative cooling. Regionally, the winds determine where there is an increase or a decrease. If the winds change little, compared to the humidity they transport, the wet regions import more water, and there could be more rain, while the dry ones could be drier [37–39].

Giorgi et al. [40] introduce a new measure of hydroclimatic intensity (HY-INT), which integrates metrics of precipitation intensity and dry spell length. They argue that the responses of these two metrics to global warming are deeply interconnected. They found clear increasing trends of HY-INT in global and regional climate models. The increase in HY-INT could be due to an increase in precipitation intensity, dry spell length, or both, depending on the region. These authors also examined late-twentieth-century observations and concluded that they also exhibit dominant positive HY-INT trends, providing a hydroclimatic signature of late-twentieth-century warming. Precipitation intensity increases because of increased atmospheric water holding capacity. However, increases in mean precipitation need compensating increases in surface evaporation rates, which are lower than for atmospheric moisture. Global warming increases potential evaporation, which, if enough moisture is available, may increase actual evapotranspiration. Potentially, there is more drying, but in drought situations, part of any extra energy goes into increasing temperatures, therefore amplifying warming over dry land. This feedback reduces the number of wet days and therefore increases mean dry spell length.

## 2. Materials and Methods

### 2.1. Study Area and Data

We analyzed precipitation data both from rain gauges and the CHIRPS database. The gauges are in 1706 sites in the whole territory of Colombia, 1062 in the Andes region. The other sites are in the Amazon, the Caribbean, the Orinoco, and the Pacific regions (respectively 77, 398, 91, and 78 stations). Data comprise daily time series of rainfall amounts. Since the method requires no missing data (Section 2.2) we trim the series to a common period between 1981 and 2013. The main reason for choosing this period is the availability of data and the compromise for the objectives of having the longest possible record and the largest number of gauges covering the whole country. Figure 1 shows the IDEAM's rain gauge network as well as the Colombian regions. The network covers a range of elevations from sea level in the Caribbean and Pacific coasts to 4150 m above sea

level in the Andes. Notice also the low density of the gauge network in the Amazon and Orinoco regions.

The Climate Hazards Group InfraRed Precipitation with Station data (CHIRPS) is a rainfall dataset at various resolutions, we use the 0.05° resolution based on (i) Climate Hazards Group Precipitation climatology, (ii) satellite imagery and (iii) in-situ station. The satellite data are infrared cold cloud duration. For Colombia, they used 3380 stations, surely with varying record length. It is, therefore, a gridded rainfall field with daily time resolution. It covers the whole country from 1981 to 2018 [8]. Funk et al. [8] present a validation of all the dataset used against rain gauges observations in Colombia for the primary rainy season (September-November) for each year. They found a correlation between CHIRPS and the average of 338 IDEAM stations of 0.97, and a mean absolute error of 38 mm. Moreover, we tested the performance of CHIRPS in Colombia in previous studies using further metrics and found satisfactory results [41,42].

Of the 37,380 CHIRPS pixels covering continental Colombia, 95.7% do not have any of the 1706 rain gauges that we used in this study, 4% have just one, and the rest (0.3%) have more than one. These figures clearly show that the rainfall gauges do not cover a considerable portion of the country. Therefore the CHIRPS dataset provide much broader space coverture for our trend analysis. Additionally, even assuming that the 3380 stations used in the CHIRPS blended gauge data were in separated CHIRPS cells, about 90% of the cells are mostly independent of gauge data considering the size of the cell and the rainfall correlation length.

We fill the missing data in the gauge records with data from the CHIRPS dataset. A condition for this filling was that the percentage of missing days for any year did not exceed 30%. Otherwise, the whole year was considered to be missing. This decision to fill missing data is due to the small number of stations that would result if any missing day dropped the whole year. Of the original IDEAM dataset, we dropped all the stations, with more than 50% missing data in the common period. Of the 1706 gauges, we filter out those having less than 30 years of complete record in the 33 years of the chosen period. The resulting base dataset has 909 rain gauges, but to test this filter criterion, we considered four other sensitivity alternatives: the first sensitivity dataset has 355 stations with no missing data in the whole record; the second sensitivity dataset has 1345 stations with at least 25 years with no missing data; the third sensitivity dataset has 1320 stations with at least 30 years of complete record in the chosen period but relaxing the condition for using CHIRPS data to fill any number of missing days; and finally, the fourth sensitivity dataset has 1629 stations with any number of full years in the period 1970–2014 and using the 30% criteria for filling gaps using CHIRPS data. These sensitivity alternatives seek to cover both broader and stricter criteria. We will see that the main results remain the same for those four sensitivity alternatives.

Colombian climate is tropical, mean annual temperature is high, above 25 °C at sea level, the diurnal range of temperature exceeds the annual range, and the annual range is minimal, less than 5 °C [10]. Precipitation is abundant compared to any other place in the world, mean annual 2830 mm over the whole country. There are places of the Pacific coast with perennial rain with mean annual totaling 12200 mm, while the average over the region is 5010 mm. Over the Orinoco and Amazon basins in Colombia, the mean annual precipitation varies from 2000 to 7000 mm per year. The average is of the order of 1500 mm on the Caribbean coast, but to the north, there are places with near 300 mm/year. In the Andean region, the mean annual precipitation ranges from 1000 to 3000 mm/year. See Supporting Information Figures S1–S7 for maps of the averages of annual precipitation and the other variables considered in this study.

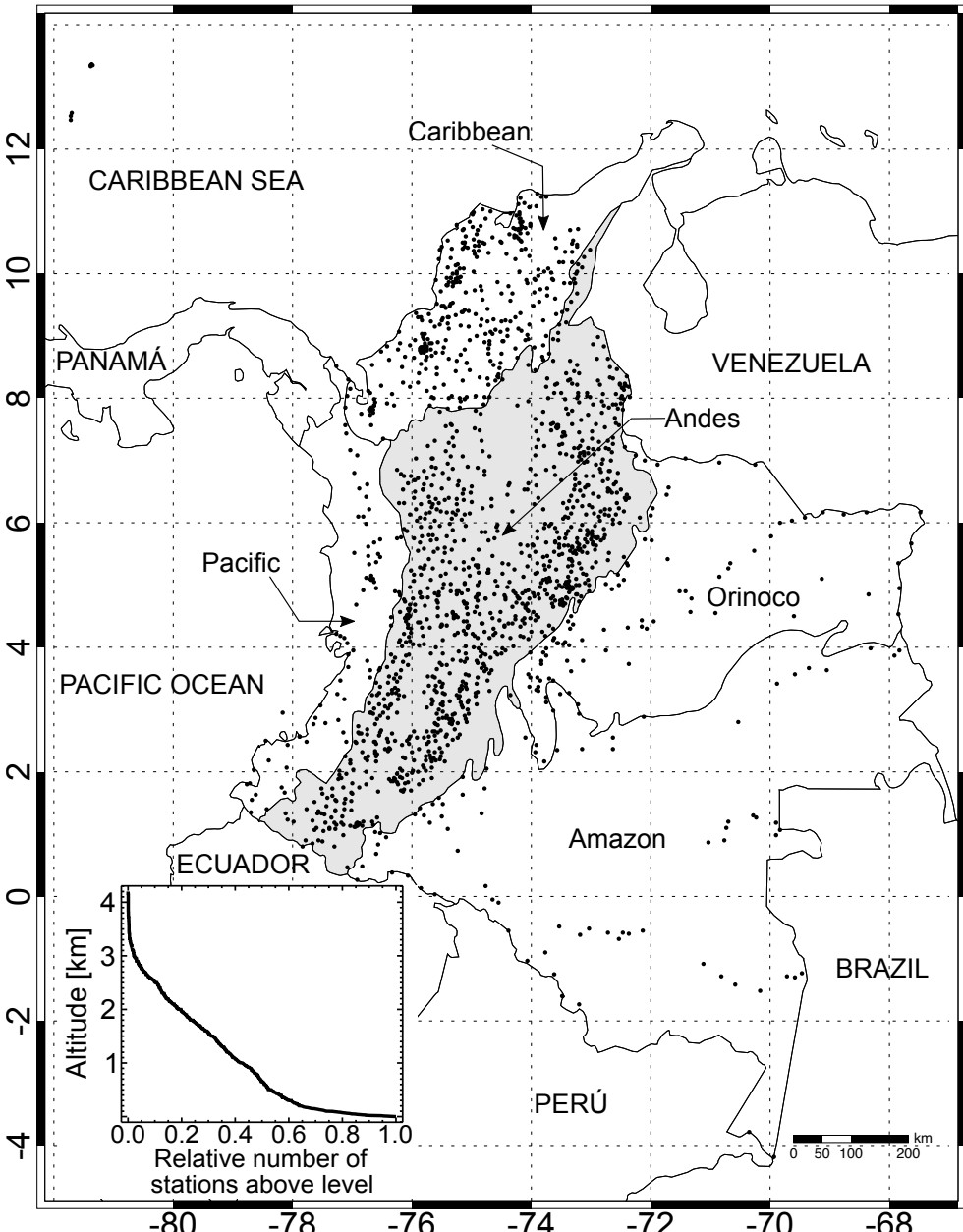

**Figure 1.** The points mark the location of the IDEAM rain gauge network used in this work. The bottom left graph shows the vertical distribution of the rain stations. The map also shows the five natural regions of Colombia [43].

Atmospheric moisture is transported toward Colombia by the trade winds from the Caribbean Sea, from the Atlantic Ocean through both the Amazon and Orinoco basins that themselves contribute with recirculating moisture. Additionally, from the Pacific Ocean, westerly winds contribute to the massive convergence of moisture over Colombia. The migration of the ITCZ and three low-level jet streams (Chocó, Caribbean, and South America) are part of the complex circulation that produces such high precipitation [44].

*2.2. Methods*

2.2.1. HY-INT Index

The following HY-INT indicator

$$\text{HY-INT} = \text{INT} \times \text{DSL} \tag{1}$$

evaluates the hydroclimatic intensity [40], where INT and DSL are the mean intensity during wet days and the mean dry spell length for each year in the record. In both cases, one works with scaled variables using the respective inter-annual mean as scale factor [40]. Therefore the long-term average of both INT and DSL is 1.

We also evaluate trends for the total annual precipitation for each year, and for WSL, the mean wet spell length in each year. We counted the number of dry days and the number of dry spells in each year. The number of wet days in a year is 365 minus the number of dry days, and therefore trends in either one are opposite. Less obvious but also true is that the number of wet runs equals the number of dry runs in a year, or they may differ by one, depending on the parity. For both reasons, we will only report trends of the corresponding dry variables. There are other relations between the variables that are worth remembering. Before scaling, INT equals P divided by the number of wet days; and DSL equals the number of dry days divided by the number of dry runs, similarly for WSL. Recall that the scaling makes possible the comparison of trends across stations or cells with different average values.

Additionally, to construct an extreme indicator of the hydrologic cycle generalizing Giorgi et al. [40] ideas, we computed the maximum daily intensity for each year (INTX) and the maximum dry spell length for each year (DSLX). Therefore, in addition to the average of the corresponding variable for each year, we take the maximum. Their product gives the HY-INTX indicator of the strength of the hydrologic cycle.

For each year in the record, we computed each one of the variables mentioned above. Therefore for each gauge and cell and each variable, we have a time series. We then proceed to evaluate the existence of trends in those time series for each gauge and cell.

### 2.2.2. Trend Analysis

We use the Mann–Kendall (MK) test [45,46] for autocorrelated data [47] to evaluate the existence of trends in the time series, and the Sen slope estimator [48] for calculating the magnitude of the trend. A summary of these techniques follows. See more details in the cited references.

The MK test null hypothesis is that the data come from independent and identically distributed random variables (iid), and hence no long-term trend exists. When the data are iid, the statistic $S$

$$S = \sum_{i=1}^{n-1} \sum_{j=i+1}^{n} \mathrm{sgn}(x_j - x_i), \tag{2}$$

has asymptotic normality with mean zero and variance

$$\mathrm{Var}[S] = \frac{n(n-1)(2n+5)}{18} - \frac{1}{18} \sum_{j=1}^{m} t_j(t_j - 1)(2t_j + 5). \tag{3}$$

In Equation (2), $n$ is the sample size, $x_t$ is the value of the time series at time $t$, and $\mathrm{sgn}(x_j - x_i)$ is defined by

$$\mathrm{sgn}(x_j - x_i) = \begin{cases} 1 & \text{if } x_j - x_i > 0, \\ 0 & \text{if } x_j - x_i = 0, \\ -1 & \text{if } x_j - x_i < 0. \end{cases} \tag{4}$$

The sum in the last term of Equation (3) accounts for the reduction in variance due to the existence of tied ranks [49]. In Equation (3), $m$ is the number of groups of tied ranks, and $t_j$ is the number of ranks in group $j$.

The standardized test statistic $Z$ is calculated by

$$
Z = \begin{cases}
\frac{S-1}{\sqrt{\text{Var}[S]}} & \text{if } S > 0, \\
0 & \text{if } S = 0, \\
\frac{S+1}{\sqrt{\text{Var}[S]}} & \text{if } S < 0.
\end{cases} \tag{5}
$$

The null hypothesis of no trend is rejected if $|Z|$ exceeds the value $|Z_{1-\alpha/2}|$ of the standard normal distribution for a given significance level $\alpha$.

The result of the MK test is sensitive to autocorrelation in the data. The existence of positive autocorrelation increases the probability of false rejection. On the other hand, negative autocorrelation increases the probability of false positive. This effect occurs because of a bias in the estimation of $\text{Var}[S]$. Hamed and Rao [47] suggested the empirical formula in Equation (6) for calculating $\text{Var}[S]$ in the presence of autocorrelation.

$$
\text{Var}[S_{ac}] = \text{Var}[S] \times \left[ 1 + \frac{2}{n(n-1)(n-2)} \sum_{i=1}^{n-1} (n-i)(n-i-1)(n-i-2)\rho_s(i) \right], \tag{6}
$$

where $\rho_s(i)$ is the autocorrelation function of the ranks of the observations.

Sen's non-parametric method [48] estimates the long-term linear trend slope of a time series as the median value of the slopes between all pairs of points in the series. For $N = n \cdot (n-1)/2$ pairs of data in the series, the $N$ slopes, $Q_i$, are calculated as shown in Equation (7). The median of the $Q_i$'s is the Sen's slope estimator.

$$
Q_i = \frac{x_j - x_k}{j - k}, \quad 1 \le j \le n-1, \quad j < k \le n, \tag{7}
$$

with $i = n - k + (j-1)(2n-j)/2 + 1$.

### 2.3. The HY-INT Trend

Even though HY-INT is not a linear function of INT and DSL, the long-term trend slope of HY-INT is a function of the trend slopes of INT and DSL. Equivalently, one can estimate it from the time series of HY-INT. Taking the time derivative of Equation (1) one obtains

$$
\frac{d\text{HY-INT}}{dt} = \text{INT}\frac{d\text{DSL}}{dt} + \text{DSL}\frac{d\text{INT}}{dt}. \tag{8}
$$

And because all the variables are scaled, what one needs is the logarithmic derivative

$$
\frac{1}{\text{HY-INT}}\frac{d\text{HY-INT}}{dt} = \frac{1}{\text{DSL}}\frac{d\text{DSL}}{dt} + \frac{1}{\text{INT}}\frac{d\text{INT}}{dt}. \tag{9}
$$

Therefore the respective temporal trend slopes, $m_{[]}$, satisfy

$$
m_{\text{HY-INT}} = m_{\text{INT}} + m_{\text{DSL}}. \tag{10}
$$

### 3. Results

Neglecting data autocorrelation in trend analysis increases the probability of error in the MK test result [50,51]. We compared the results of the classic MK test and the MK test for autocorrelated data proposed by Hamed and Rao [47] in our 1629 series dataset. We found that ignoring the autocorrelation may lead to false trends of the order of 20% of the gauges and false no trends of the order of 10% of the gauges.

Table S1 presents in each row the four elements of the confusion matrix for the MK test that does not take into account autocorrelation in comparison with the one that does. We take this last one as the correct method for the comparison. The largest error (from 18% to 25%) comes from false trends. However, there are also errors due to false no trends (from 11% to 15%). As a result, the accuracy (total number of hits) is between 60% and 71%.

Thus, the recommendation is to take the series' autocorrelation into account to evaluate the significance of trends.

We also considered the possible implication of the definition of the starting date of the year. Moreover, the calendar year, we considered a hydrology year starting on April 1st. The idea was that the end of the calendar year might split the most extended dry spell, because the dry season usually starts in mid-December and ends in March in Colombia. However, we found no statistically significant difference depending on the anthropic definition of the year.

Figure S8 illustrates two of the 909 cases of the trend analysis. Notice the treatment of the missing years that may come for any missing day. For the Susacón gauge in the left part of the Figure, the trends in P, INT, and HY-INT are decreasing and statistically significant. However, the trend in DSL is not. The right side of the Figure is for La Línea El Porvenir rain gauge. One can see that DSL, INT, and HY-INT variables have significant positive trends. In contrast, P has an increasing non statistically significant trend.

Table 1 summarizes the results of the trend analysis for the more relevant variables computed for the 909 rain gauges in the base dataset, and the 37,012 cells in CHIRPS dataset. Figures 2–9 display the results.

The first observation is that only a minority of the rain gauges show significant trends for any of the variables. Among the variables, INT shows the largest percentage of significant trends, but only reaching 21% of the stations. The least percentage is for the variable DSLX with only 10%. Similarly, a small percentage of all the CHIRPS cells show significant trends for any of the variables. Again, for the intensity (INT), the percentage is one of the largest, reaching only around 25% of the cells. The other is for the mean wet spell length (WSL), which has a similar percentage of significant cells. The lowest percentage of significant cells is for the maximum dry spell length (DSLX), with only 5%. Summarizing, the majority of the stations or cells do not show significant trends for any of the variables analyzed.

Figure 10 presents the histograms of the HY-INT, INT, and DSL trend slopes. In all the cases the histograms show that the majority of stations do not have significant trends.

**Table 1.** Basic statistical analysis for the trends in base dataset (909 stations) and in CHIRPS dataset (37012 cells covering Colombia). Columns show percentage of significant trends (SIG) and percentage of increasing trends as a proportion of the significant ones (INC). Results for the following variables: P: Total Annual Precipitation, INT: averaged scaled intensity on wet days, DSL: averaged scaled dry run length; HYI-NT = INT× DSL; N stands for the number; WSL: averaged scaled length of wet runs; INTX: the maximum daily intensity; DSLX: the maximum dry run length; HY-INTX = INTX× DSLX; and WSLX: the maximum wet run length.

| Variable | % SIG | % INC | % SIG | % INC |
|---|---|---|---|---|
| | **Rain Gauges** | | **CHIRPS** | |
| P | 13 | 79 | 10 | 76 |
| INT | 21 | 54 | 25 | 48 |
| DSL | 17 | 20 | 10 | 48 |
| HY-INT | 21 | 33 | 23 | 38 |
| N Wet Days | 18 | 80 | 20 | 32 |
| WSL | 19 | 74 | 25 | 22 |
| N Wet Runs | 15 | 66 | 13 | 94 |
| INTX | 10 | 62 | 13 | 40 |
| DSLX | 10 | 31 | 5 | 47 |
| HY-INTX | 10 | 38 | 9 | 48 |
| WSLX | 14 | 74 | 18 | 16 |

Another clear observation is that the extreme variables do not show a larger percentage of significant trends in comparison with the corresponding average variables: For INTX, the percentage of rain gauges is 10, compared with 21 for INT. Similarly, for DSLX,

the percentage is 10, in comparison with 17 for DSL. For HY-INTX, the percentage is 10, whereas, for HY-INT, it is 21. The same picture applies to the CHIRPS data, where the percentage of significant trends for the extreme variables is half the corresponding for average variables, except for WSLX, where the ratio is 18 to 25. Summarizing, statistical significant trends in extreme variables are less frequent than in average ones.

The analysis of increasing and decreasing trends among the stations or cells with statistically significant trends is interesting (see Table 1 and Figure 10). There is a majority of increasing trends both for rain gauges and CHIRPS cells for P, annual precipitation, and for the number of wet and dry runs; with a closer agreement between datasets for P, and less agreement for the number of runs where the corresponding positive trend percentages are 66 and 94. There is also accordance between rain gauges and CHIRPS for the majority of decreasing trends for HY-INT, HY-INTX, DSL and DSLX. For the rest of the variables the prevalence is opposite between the two datasets (INT: 54 vs. 48, number of wet days 80 vs. 32, INTX: 62 vs. 40, WSL: 74 vs. 22, and WSLX: 74 vs. 16). In all these last variables, the majority of significant trends are positive for rain gauges and negative for CHIRPS. It is necessary to complement this previous analysis of the whole country with a regional zoom. Table S3 presents a summary of the regional trends. Of the above countrywide tendencies, only the increasing ones for P and NWR remain valid for all the regions. The decreasing trends for HY-INT, HY-INTX, and DSLX do not hold for all the regions, nor is there agreement between the two datasets.

Part of the explanation of this discrepancy is related to the space coverage of the two datasets. There are very few rain gauges in the eastern part of the country, but CHIRPS has total coverage. As a result, CHIRPS shows a dipole in those variables' trends, not observed in the rain gauges. Figures 3, 5, 7 and S10 illustrate this dipole. However, this difference in space coverage is not the whole story because trends in the Andes region, with an adequate number of gauges, have different signs between the two datasets. Whereas most statically significant Andean rain gauge trends for INT, DSL, HY-INT, DSLX, and HY-INTX are decreasing, they increase for CHIRPS. The differences are ample: for DSL, the percentage of positive trends are 23 vs. 81; for the number of wet days, 85 vs. 2; and for WSL, 82 vs. 2.

An entirely satisfactory explanation of this discrepancy for the Andes region remains open. Among the observations that may play a role, we indicate the following: (i) For almost all the variables, the percentage of statistically significant trends is larger for the CHIRPS database than for the gauges. For INT, the figures go from 20 to 34, and from 18 to 46 for WSL. (ii) The space distribution of the small number of rain gauges with significant trends does not seem to show any pattern. They are interspacing with the majority of nonsignificant trends. (iii) In contrast, there is a clear pattern for the spatial distribution of trends in the CHIRPS. (iv) The satellite infrared cold cloud duration method of CHIRPS may have problems for regions with this kind of complex topography. Nevertheless, our validation is satisfactory [41,42].

Only 33% of the stations and 38% of the cells with significant trends show positive trends for HY-INT. The analysis for the two databases is different. First for CHIRPS both of the factors of HY-INT, INT and DSL, have positive trends for approximately half among the significant cells. Clearly, there is an east-west dipole for the three variables (ses Figures 3, 5 and 7), with decreasing trends in the east and north and increasing in the west, with high percentages in all the cases. Therefore for the Pacific and Andes region of Colombia's humid climate HY-INT, the indicator proposed by Giorgi et al. [40] to measure the strength of the hydrologic cycle, makes sense. For those two regions, the increasing hydroclimatic intensity as an integrated response to global warming implies increasing risks for systems that are sensitive to wet and dry extremes. The explanation of the negative HY-INT trends for the Caribbean, Orinoco and Amazon regions of Colombia comes from the decreasing trends in intensity and dry spell length, despite increasing total annual precipitation, associated with more rainy days and more number of runs.

For the rain gauges the story is different. Despite the tendency for INT to increase, the explanation for the low percentage of significant HY-INT gauges with positive trends is the sign of the trends in DSL, 80% have negative trends among significant DSL trends.

Figure 11 allows further analysis of the result about HY-INT. Notice that almost all stations and cells with positive trend slope for HY-INT have positive trend slopes for both INT and DSL. Conversely, almost all with negative trends for HY-INT have negative trends for INT and DSL. Another interesting observation from the figure is the difference in the dispersion of the points between the two datasets. Because CHIRPS data are cells of finite size, there is significant smoothing compared with rain gauges that are point observations and therefore capture the natural irregularity of the rainfall field.

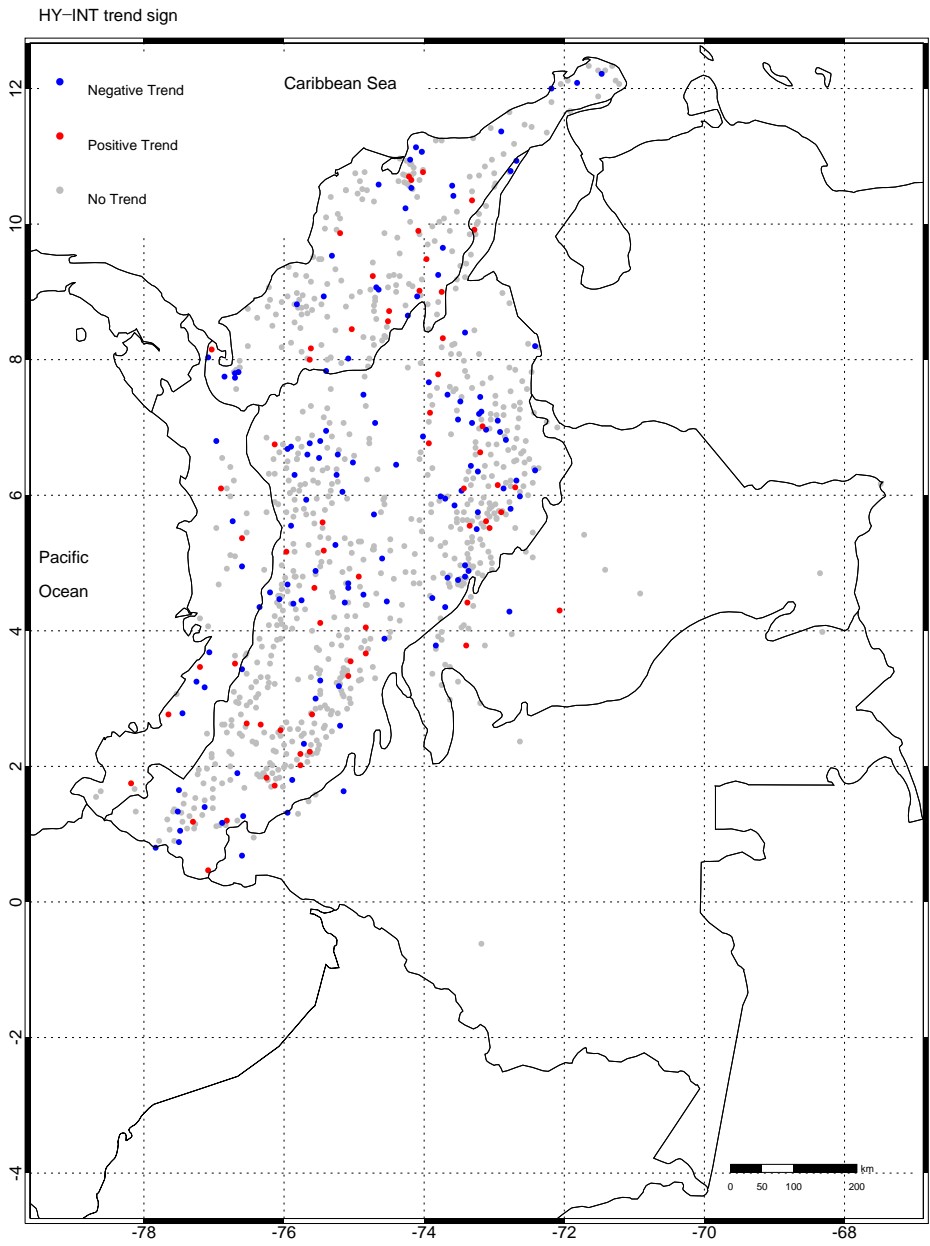

**Figure 2.** Map of the trend sign for HY-INT for rain gauge base dataset. Non-significant trends are plotted in gray.

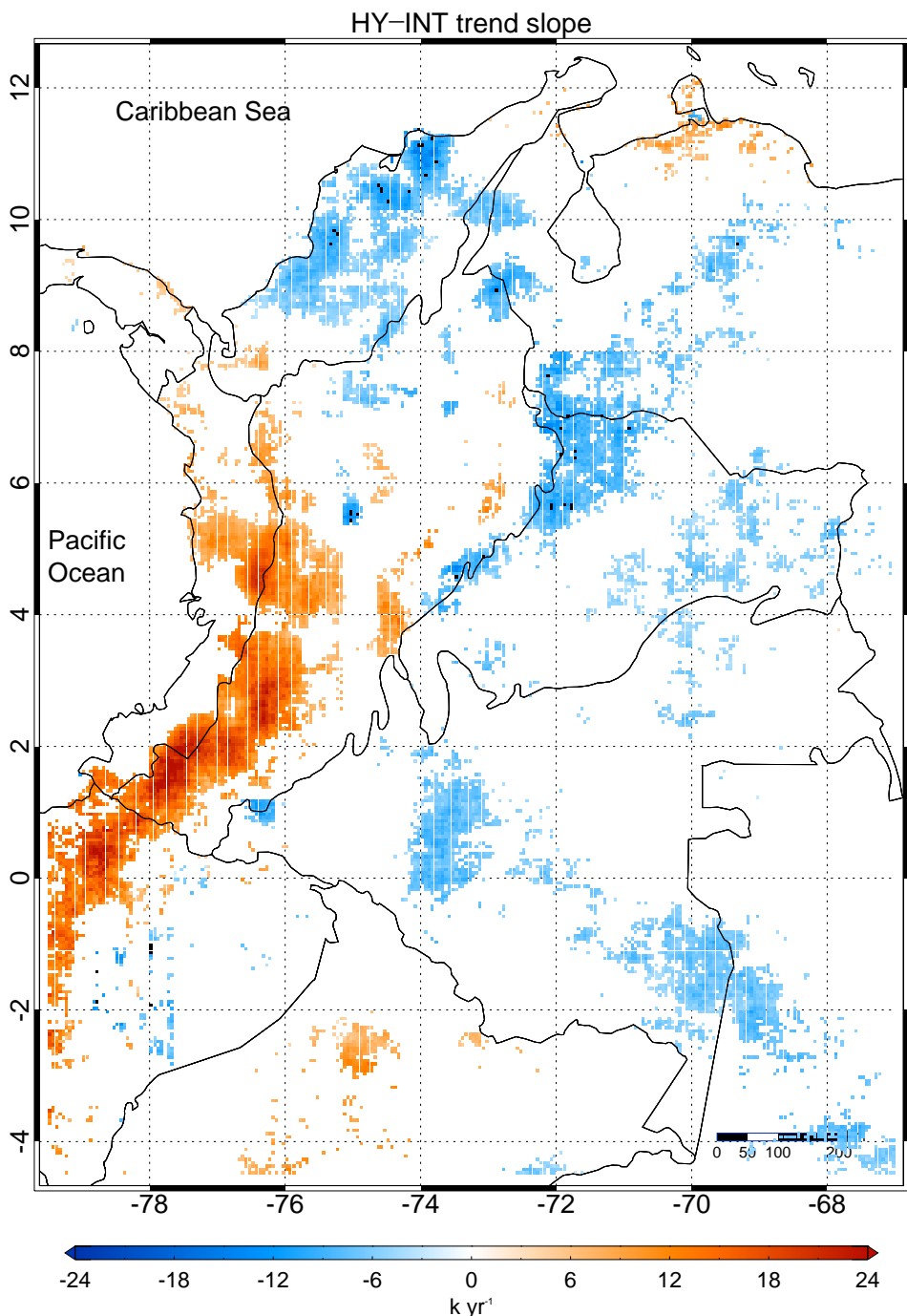

**Figure 3.** Map of the trend slope for HY-INT for CHIRPS dataset. Non-significant trends are not plotted. Color scale in $\mathrm{k\,yr^{-1}}$.

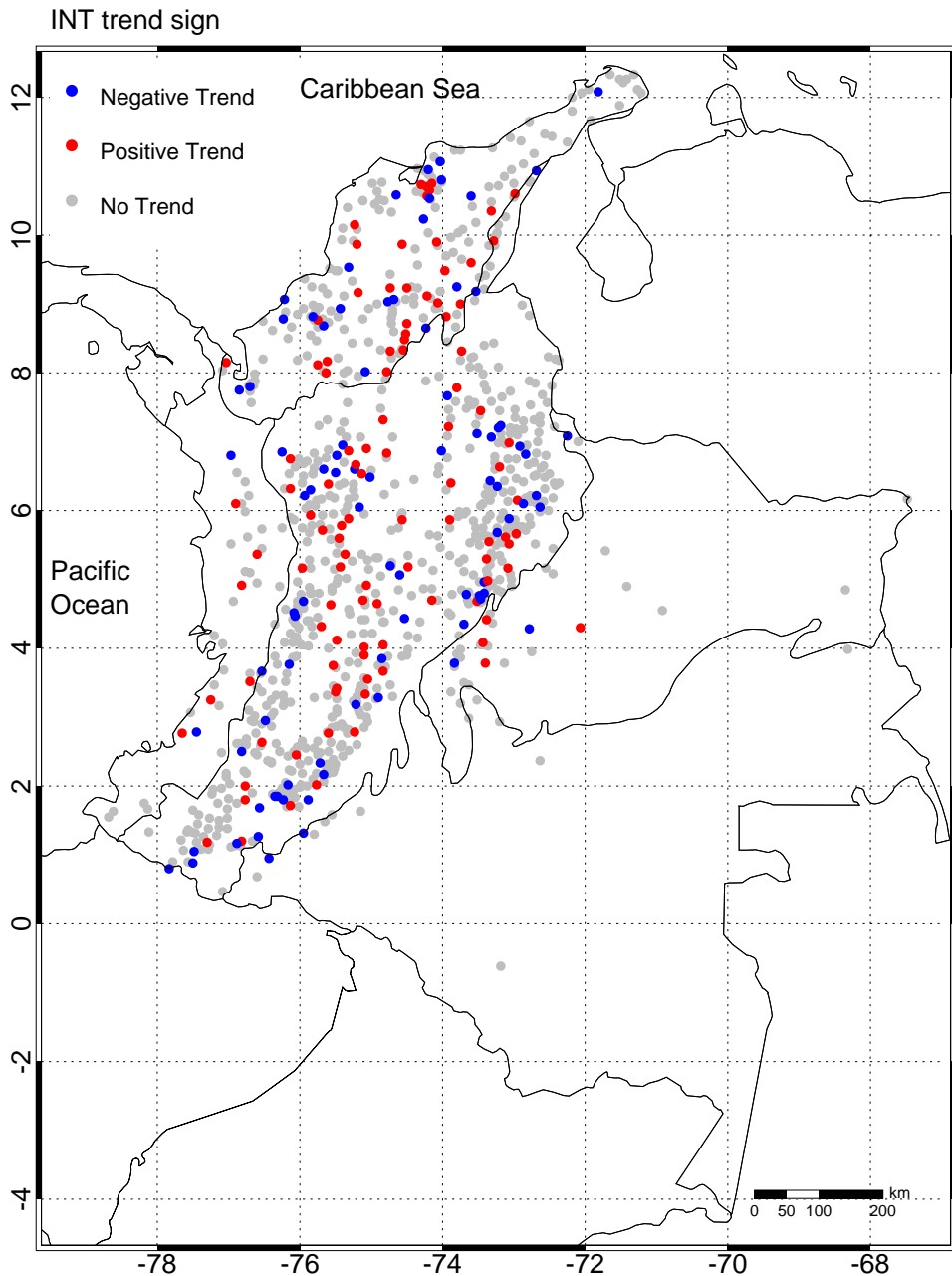

**Figure 4.** Map of the trend sign for INT for rain gauge base dataset. Non-significant trends are plotted in gray.

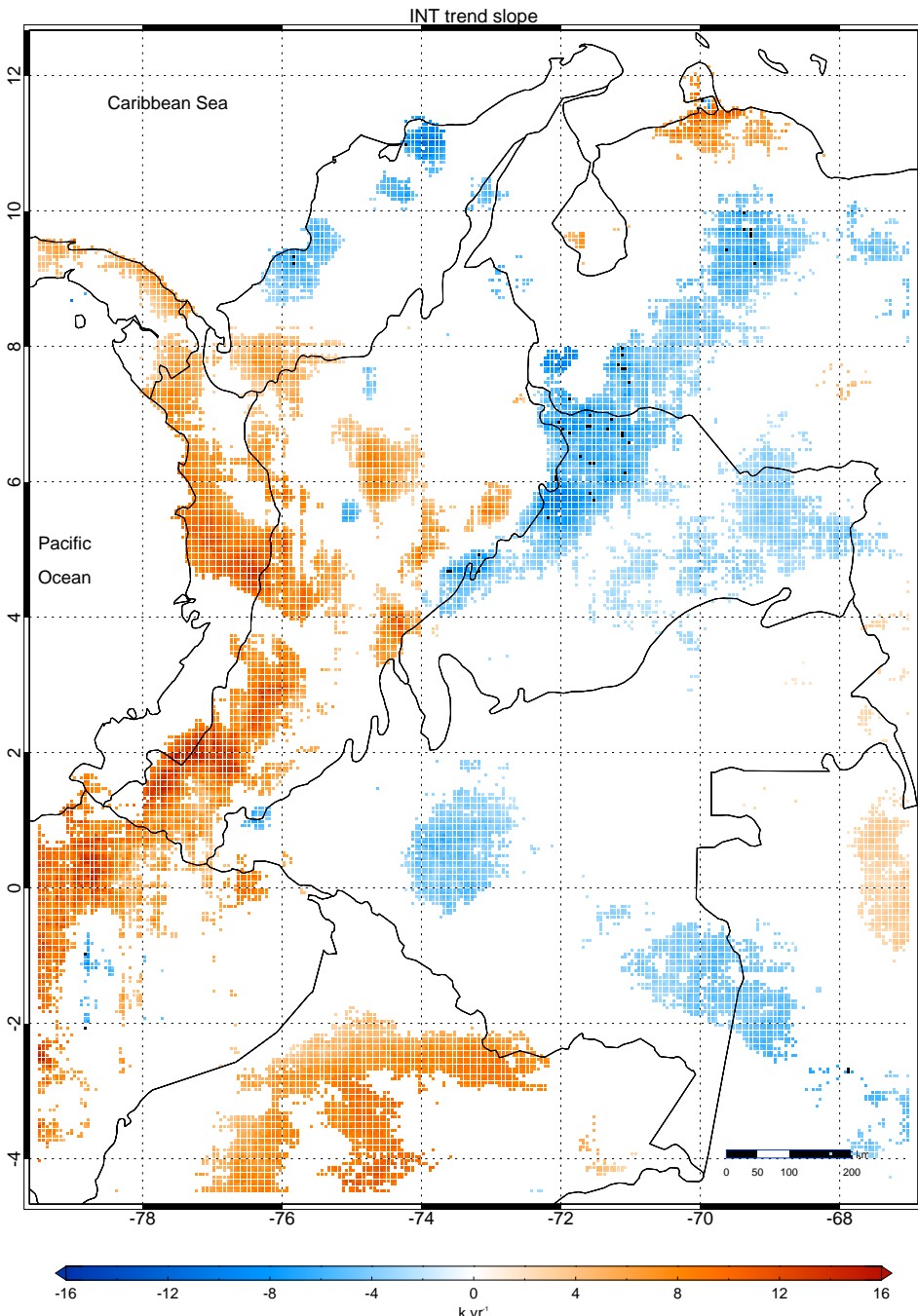

**Figure 5.** Map of the trend slope for INT for CHIRPS dataset. Non-significant trends are not plotted. Color scale in $k\,yr^{-1}$.

Figure S9 shows the histograms of the trend slopes for the annual precipitation, the number of wet days, and the number of wet spells. Again, the majority of the stations do not have significant trends. However, among the significant ones, there is a majority of positive trends.

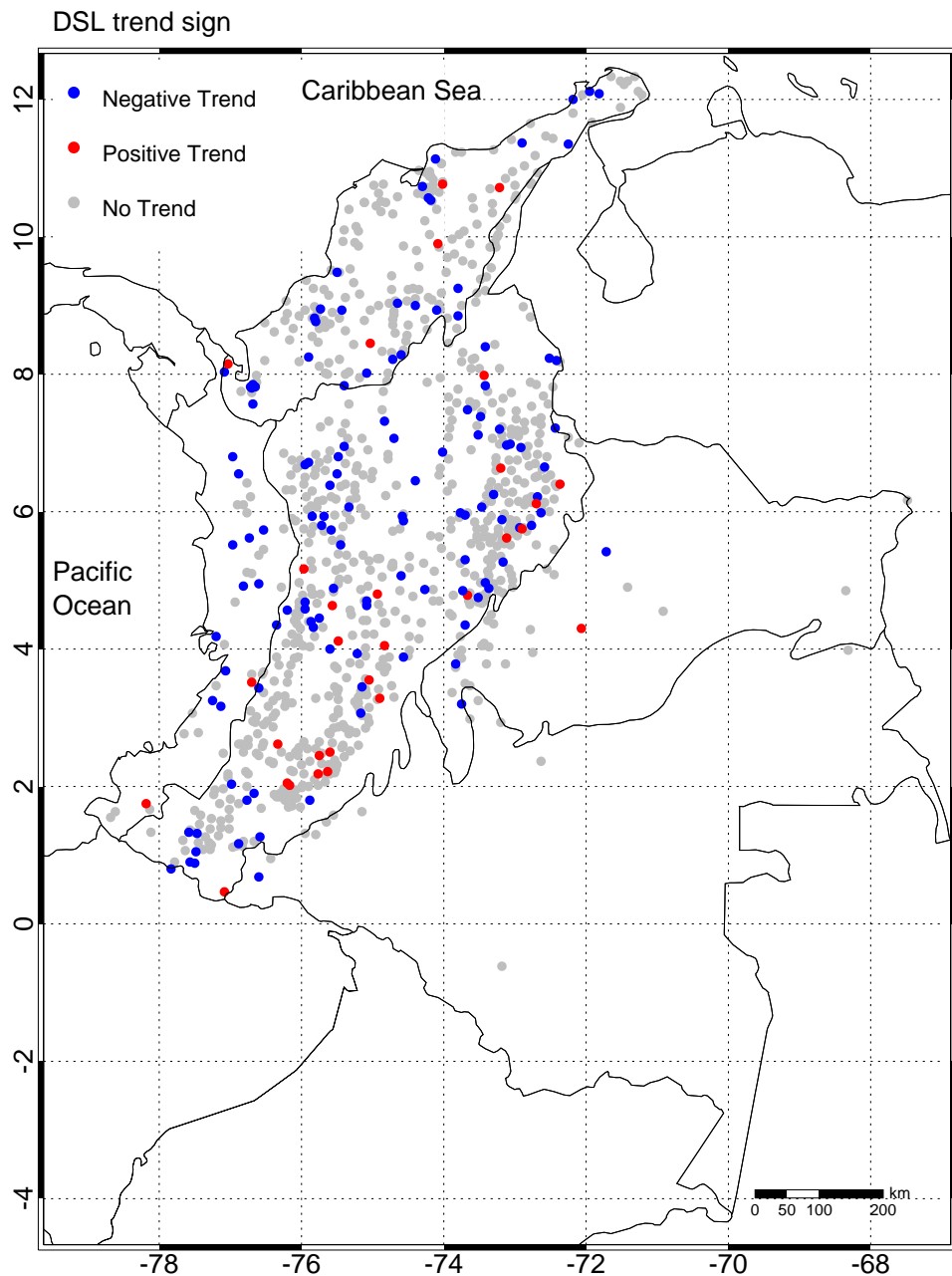

**Figure 6.** Map of the trend sign for DSL for rain gauge base dataset. Non-significant trends are plotted in gray.

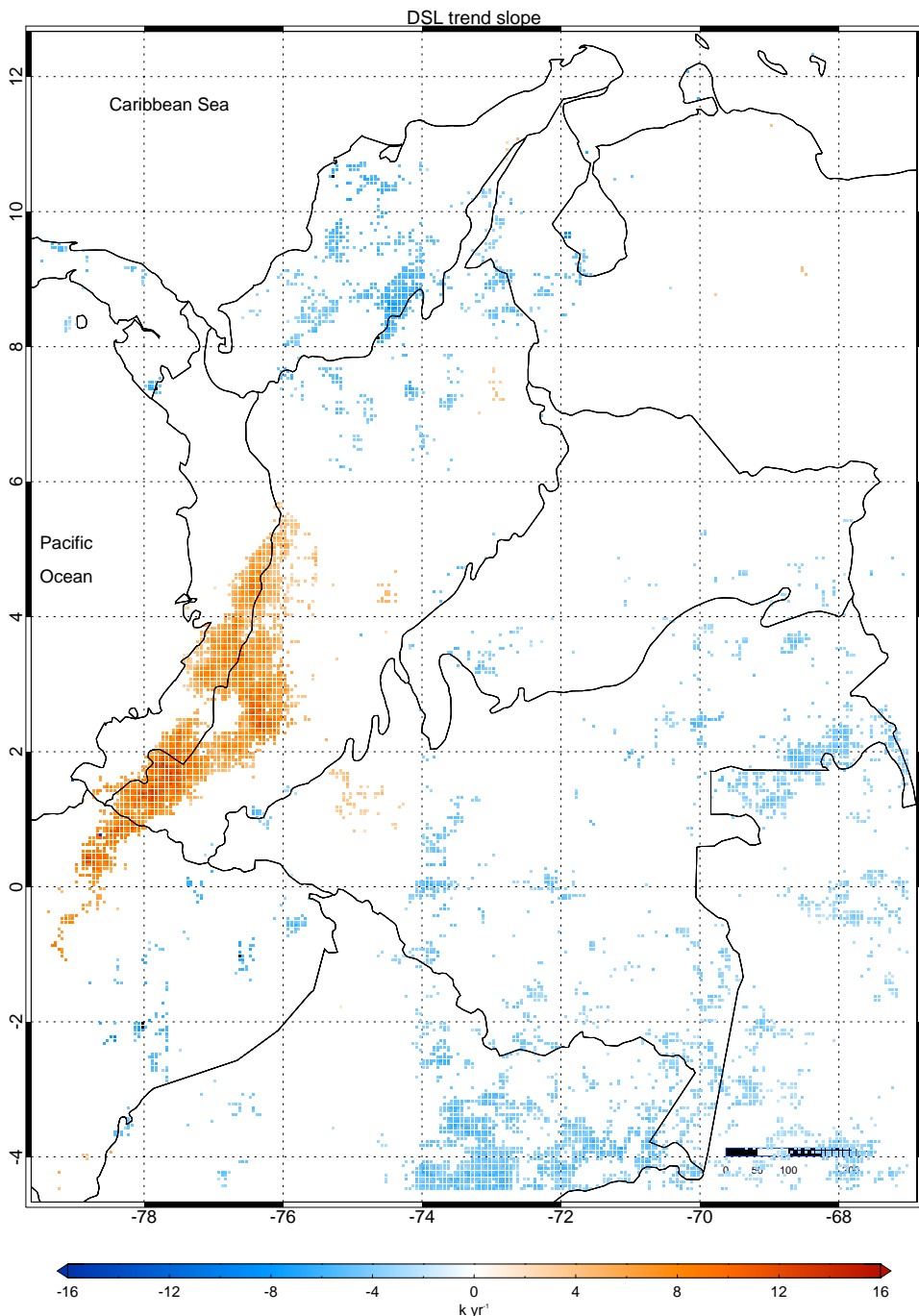

**Figure 7.** Map of the trend slope for DSL for CHIRPS dataset. Non-significant trends are not plotted for CHIRPS. Color scale in $k\,yr^{-1}$.

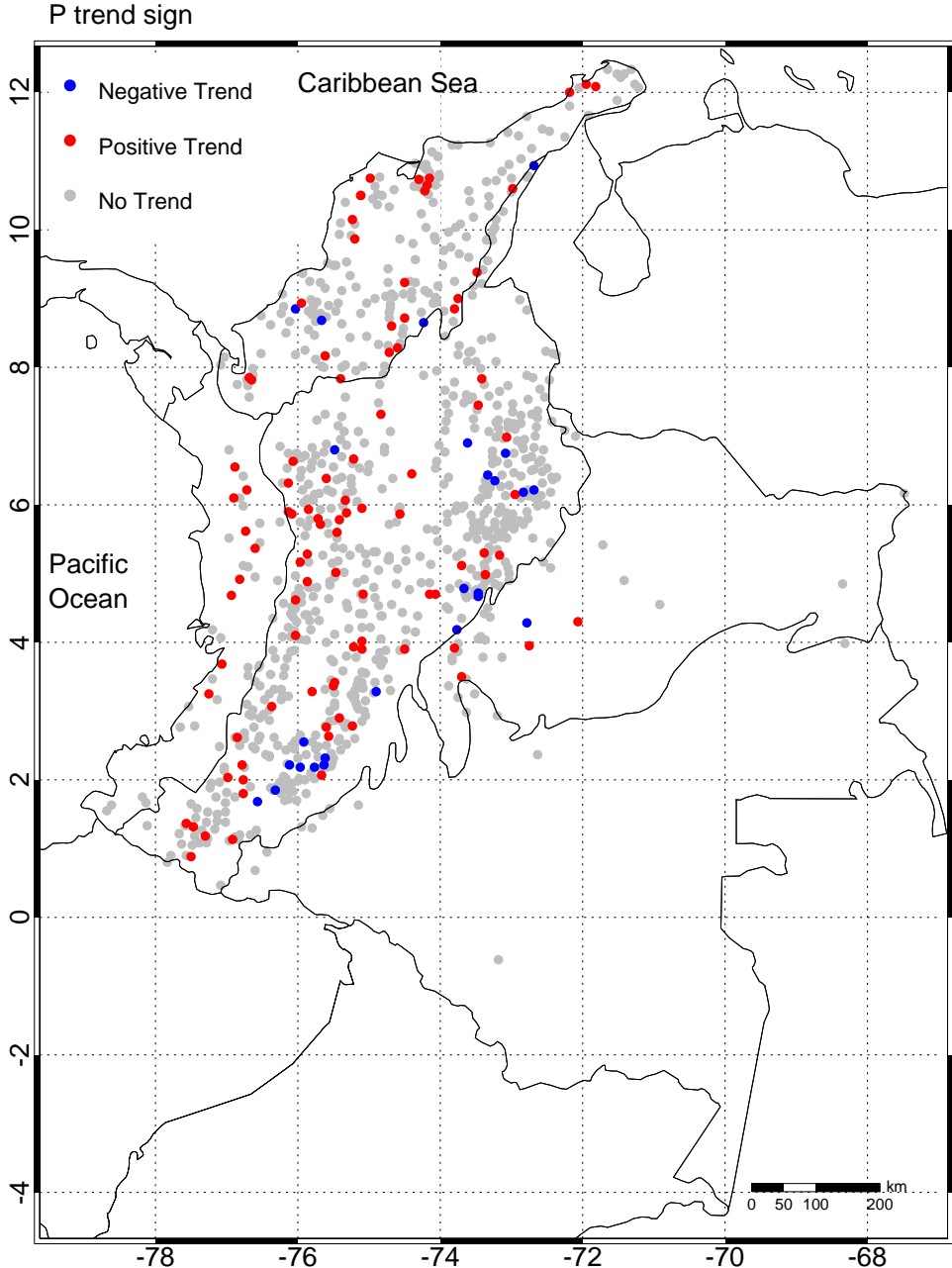

**Figure 8.** Map of the trend sign for P for rain gauge base dataset. Non-significant trends are plotted in gray.

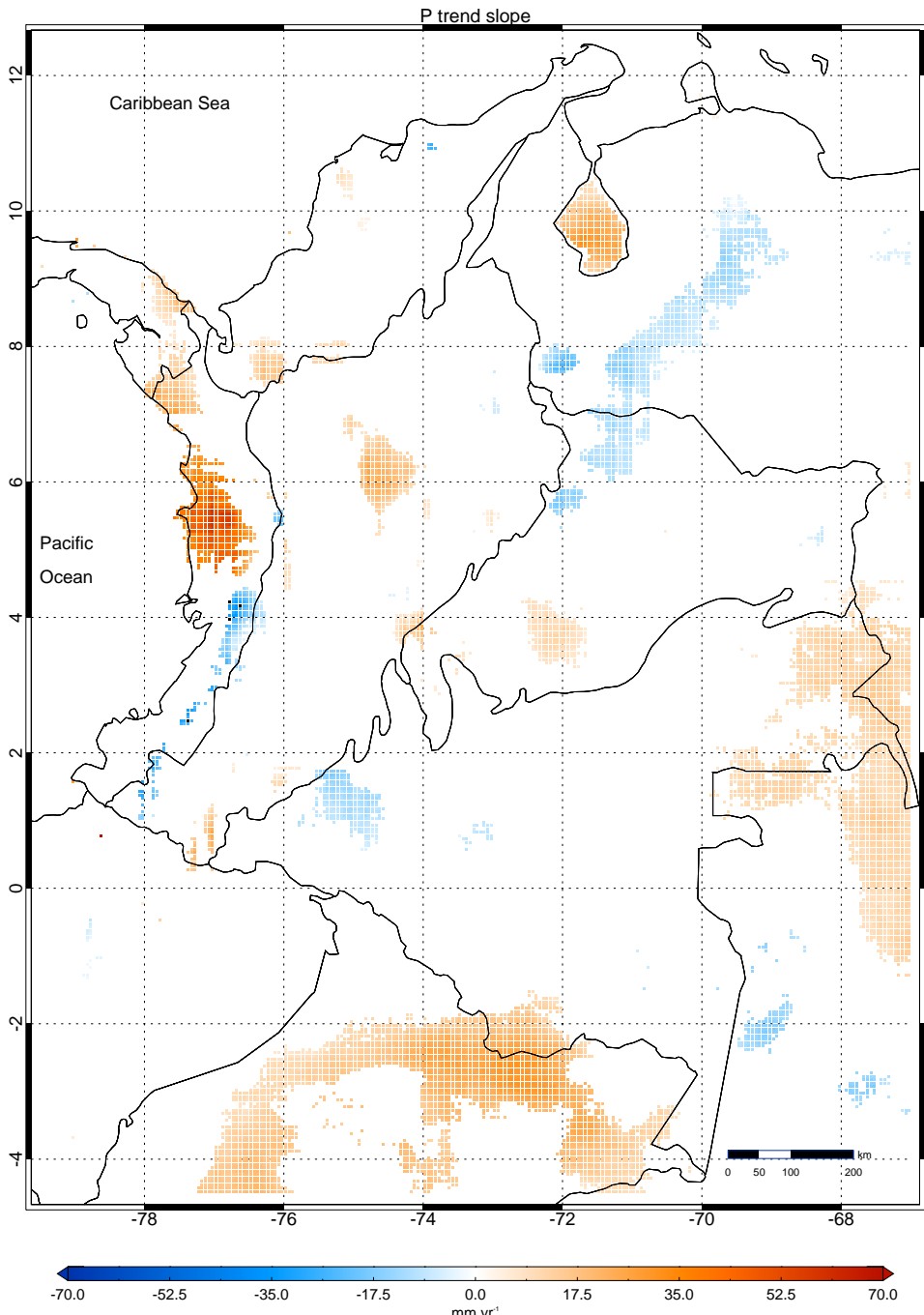

**Figure 9.** Map of the trend slope for P CHIRPS dataset. Non-significant trends are not plotted for CHIRPS. Color scale in mm/yr.

　　Table S2 summarizes the sensitivity analysis for the different alternatives considered for selecting the rain gauges. The overall conclusion is that the main results are consistent among the various datasets. The base dataset is in between the alternatives. Changes in the percentage of stations with significant trend for all the variables are relatively small, less than 3 points in 20. For some variables, the percentage of increasing trends among the significant trends does change more. For instance, the differences are somewhat more significant for the variables DSL, the number of dry days, WSL, DSLX, HY-INTX, and INT. In general, the fourth alternative is the one with more different percentages, whereas the other three and the base dataset are close together. Recalling that the fourth alternative

dataset consists of 1629 stations without consideration for the record length, one can disregard it, although it follows the general tendency of the results.

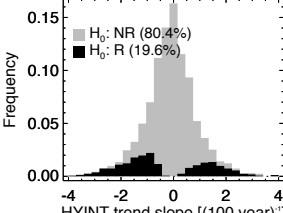 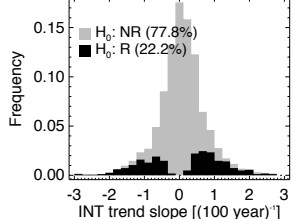 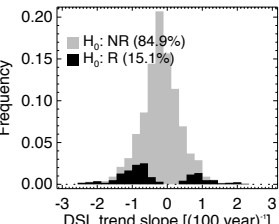

**Figure 10.** Histograms of the HY-INT (**left**), INT (**center**), and DSL (**right**) trend slopes of the fourth sensitivity rain gauge dataset. Non-significant trends in grey and significant trends in black. Results are similar for other datasets.

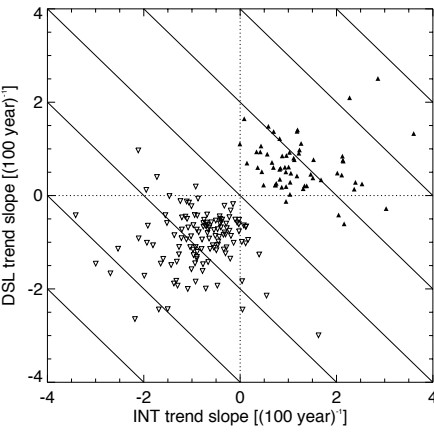 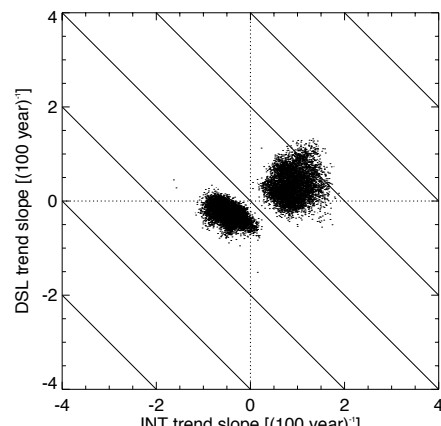

**Figure 11.** Dispersion diagram of the DSL trend slope vs. INT trend slope for all stations in the base dataset (**left**) and all pixels in the CHIRPS dataset (**right**) with significant HY-INT trend slope. Notice that because of Equation (10), the trend slope of HY-INT is the sum of the trend slopes of INT and DSL. This equation explains the slanted iso-lines for the HY-INT trend slope.

Finally, we looked into the results for the existence of associations between the trend slopes and geographical characteristics (latitude, longitude, elevation) or the type of annual rainfall regime [15]. No clear pattern was found in any of the cases.

## 4. Discussion and Conclusions

The East-West dipole in the CHIRPS observed trends for the HY-INT index, intensity, the number of wet days, and the length of wet runs suggest a climate change pattern that needs confirmation and deserves interpretation. Changes in ENSO impact the west part of the country more directly, whereas the east part is more related to the Atlantic or Amazon basin, where other processes may be developing, see for instance Sun et al. [52], Lambert et al. [3], Betts et al. [7] or Pietschnig et al. [53].

The issue of changes in ENSO due to climate change is a significant area of debate [54]. Some argue that ENSO could become more frequent and intense with global warming, but others argue that it could become weaker or more located on the central rather than the eastern Pacific. The effect of canonical ENSO over Colombia is to produce dryer weather in most of the country, but a more central ENSO effect is not as clear [6]. Therefore the issue is very relevant to elucidate the effects of climate change over Colombian precipitation. In that sense, one might speculate that our results of a majority of increasing annual precipitation trends among the significant ones for the Pacific region is consistent with the nonlinear ENSO warming suppression and possible strengthening of the Chocó jet.

Our results about the trends in annual precipitation agree in some way with the previous studies reported in Section 1.1 that considered rain gauges: Increasing trends prevail over decreasing trends among the statistically significant ones. However, we found a large number of stations and cells with non-significant trends.

One result of this work is that neither the existing records of precipitation in the rain gauge network of Colombia nor the CHIRPS dataset shows a clear signal of statistically significant trends. Only approximately 20% of the gauges or cells present significant trends. This observation is valid for all the variables we studied, even with a lower percentage of significant trends for some.

The humid tropical climate of Colombia is probably a factor for explaining the scarceness of significant trends and the sign of the found trends. Stevens and Bony [55] argue that changes in precipitation require changes in global circulation. The sole increase in absolute moisture is insufficient to change precipitation, at least in an average sense. In that sense, there are no reports of changes in the trade winds or the low-level jets that advect moisture to Colombia, except for the Chocó jet. This intensification of the jet is consistent with the Pacific region showing a predominant tendency to become wetter.

The lack of a space pattern for the trends in the rain gauges dataset in the Andes is another striking evidence. This lack of patterns contrasts with the results coming from the CHIRPS dataset. The irregularity of the precipitation field probably plays a role because rain gauges sample the field at points. In contrast, a CHIRPS data cell corresponds to a space average in a relatively large area (31 km$^2$).

Another result is about the HY-INT index of Giorgi et al. [40] to quantify the hydrologic cycle's intensity. For many parts of the globe, it may be true that rainfall intensity and dry spell length are deeply interconnected. However, the trends in rain gauge observations suggest otherwise. At least for the few gauges with significant trends, the two factors in the definition of HY-INT, rainfall intensity, and dry spell length, do not necessarily go together. For instance, 54% (104 out of 191) of the station with significant INT trends have a positive trend. However, of the 104, only 45% have positive DSL trends.

Nevertheless, trends in CHIRPS observations do follow the global trends reported by Giorgi et al. [40]. For 21% of CHIRPS cells with significant trends, 62% have negative trends, mostly in the Caribbean and the eastern regions of Colombia, the Amazon, and the Orinoco. There, both INT and DSL have predominantly decreasing trends. Moreover, there are increasing trends in the western part, the Pacific and Andes regions, with increasing components. This observation points to the direction mentioned above about the dipole, the western part of the country, with a more intense hydrological cycle.

We complemented HY-INT, the indicator of the hydrologic cycle's intensity, by defining an extreme version, HY-INTX, the product of the maximum daily rainfall times the maximum dry spell length. The dipole is similar to HY-INT, increasing western Colombia trends and decreasing ones to the east and north. However, statistically significant trends in extreme variables are less frequent than in average ones.

**Supplementary Materials:** The following are available online at https://www.mdpi.com/article/10.3390/cli9070120/s1, Figure S1: Mean annual precipitation over the period 1981–2018 from the CHIRPS database (left). Mean annual number of rainy days over the same period (right); Figure S2: Average HY-INT over the period 1981–2018 from the CHIRPS database (left). Average HY-INTX over the same period (right); Figure S3: Mean average intensity over the period 1981–2018 from the CHIRPS database (left). Mean maximum intensity over the same period (right); Figure S4: Mean wet run length over the period 1981–2018 from the CHIRPS database (left). Mean maximum wet run length over the same period (right); Figure S5: Mean dry run length over the period 1981–2018 from the CHIRPS database (left). Mean maximum dry run length over the same period (right); Figure S6: Mean number of runs over the period 1981–2018 from the CHIRPS database; Figure S7: Histograms of mean annual precipitation over the period 1981–2018 from the CHIRPS database (left) and mean annual number of rainy days over the same period (right); Figure S8: Two examples of trend analysis for (top to bottom) P, DSL, INT, and HY-INT for two representative stations, left panel: Susacón in Boyacá, at 2550 m.a.s.l., right panel: La Línea El Porvenir in Risaralda, at 1955 m.a.s.l.; Figure S9:

Same as Figure 2 for the trend slope of the total annual precipitation (left), the number of wet days in the year (center) and the number of wet spells in the year (right); Figure S10: Maps of the trend slope for the number of wet days (left) and the number of wet runs (right) for the CHIRPS dataset. Non-significant trends are not plotted; Figure S11: Maps of the trend slope for the average length of wet runs (left) and the maximum length of wet runs (right) for the CHIRPS dataset. Non-significant trends are not plotted; Table S1: Confusion matrices for the evaluation method of the significance of the trends for each of the indicated variables without taking into account autocorrelation; Table S2: Comparison of sensitivity alternatives for data filtering. Percentage of stations with a significant trend for each of the four sensitivity datasets (%S1 to %S4) and percentage of those with positive trends (%P1 to %P4). Variable symbols are the same as in Table 1.

**Author Contributions:** Conceptualization, O.M. and A.O.; methodology, O.M., V.U. and A.O.; software, O.M., V.U. and A.O.; validation, O.M., V.U. and A.O.; formal analysis, O.M., V.U. and A.O.; writing—original draft preparation, O.M., V.U. and A.O.; writing—review and editing, O.M., V.U. and A.O.; visualization, O.M., V.U. and A.O.; supervision, O.M.; project administration, O.M., V.U. and A.O.; funding acquisition, A.O. All authors have read and agreed to the published version of the manuscript.

**Funding:** This work was supported by Minciencias, Universidad Nacional de Colombia, Universidad EIA and Interconexión Eléctrica S.A. under grant contract number 80740-540-2020.

**Data Availability Statement:** Datasets from IDEAM are available for discharge from their web page http://dhime.ideam.gov.co/atencionciudadano/ accessed on 4 October 2019. CHIRPS datasets are available in [8], also at https://www.chc.ucsb.edu/data/chirps accessed on 18 February 2019.

**Acknowledgments:** Support from Universidad Nacional de Colombia is greatly acknowledged. IDEAM, the Colombian Institute for Environmental Studies provided the rain gauge records. CHIRPS data-set was produced by the Climate Hazards Group and the U.S. Geological Survey (USGS), with support from the U.S. Agency for International Development (USAID), the National Aeronautics and Space Administration (NASA), and the National Oceanic and Atmospheric Administration (NOAA).

**Conflicts of Interest:** The authors declare no conflict of interest. The funders had no role in the design of the study; in the collection, analyses, or interpretation of data; in the writing of the manuscript, or in the decision to publish the results.

## Abbreviations

The following abbreviations are used in this manuscript:

| | |
|---|---|
| P | Total annual precipitation. |
| Number of rainy days | Number of days with precipitation equal or larger than 1 mm. |
| Number of dry days | Number of days with precipitation less than 1 mm. |
| NWR | Number of wet runs |
| NDR | Number of dry runs |
| INT | Average intensity = P/Number of rainy days. |
| DSL | Dry Spell Length = Average length of dry runs |
| HY-INT | INT×DSL |
| WSL | Wet Spell Length = Average length of wet runs |
| INTX | Maximun intensity = Maximum daily rain |
| DSLX | Dry Spell Length = Maximum length of dry runs |
| HY-INTX | INTX×DSLX |
| WSLX | Wet Spell Length = Maximum length of wet runs |
| CHIRPS | Precipitation dataset selected for the Colombian territory from Climate Hazards group Infrared Precipitation with Station data |
| ENSO | El Niño-Southern Oscillation |
| IDEAM | Instituto de Hidrología, Meteorología y Estudios Ambientales |
| ITCZ | Inter-Tropical Convergence Zone |
| MK | Mann-Kendall |

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
