# Peer review of "Trends of Hydroclimatic Intensity in Colombia"

_climate, doi:10.3390/cli9070120_

Round 1

Reviewer 1 Report

The study from Mesa et al. investigates trends in different precipitation variables using observations and CHIRPS. The manuscript is mostly well written but its organization and the discussion of the results must be improved. I think the title is adequate. I believe that the methodological treatment is adequate.

I think that in the present form this manuscript needs major revisions before publication. Therefore, I suggest the authors to resubmit the manuscript, advising them to provide effective responses to the issues here evidenced.

Major Comments

  • General comments: a) The graphic art of the figures needs to be improved. Scales and text are not easy to read. b) To improve the presentation and discussion, I suggest including in the manuscript some of the figures that are in the annexes, such as: Fig A10 and A11. It may facilitate discussion of the results that are now weakly exposed.

  • The objective of the paper must be clearly stated in the last paragraph of the introduction. I suggest avoiding some general statements like: The focus of this work is the impact of climate change on precipitation. This study does not invetigate the attribution of precipitation trends to climate change, nor does it evaluate their impacts.

  • The results in Table 1 are confusingly stated. Please simplify the notation and clearly indicate when there is a trend or not and when it is significant or not.

  • Again, authors state: “The analysis of positive and negative trends among the stations or cells with statistically significant trends is interesting (Table 1)”. I cannot find this analysis in Table 1. Or if it is, it is not exposed correctly. In addition, authors say: “For the rest of the variables (INT, DSL, number of wet days, INTX, WSL, and WSLX), the prevalence is opposite between the two data sets. In all these last variables, the significant trends are positive for rain gauges and negative for CHIRPS”. Where do you see that?

  • The improvement in the presentation and discussion of the results will allow the authors to obtain more precise conclusions. “For example, the authors state that: In that sense, our results suggest that ENSO is not becoming more intense or frequent, but they are in accordance with increasing frequency of central events”. The paper does not study relationships of trends in local precipitation with ENSO, nor does it make attributions of trends to climate change. Please draw conclusions and discussions according to your findings. I understand that you can improve the discussion with global forcing analysis or other previous works that evaluate impacts of climate change, but you should refer to your findings and discuss them with previous ones, without confusing the reader. Along these lines, the introduction and the conclusions diverge from the content of the work itself.

Minor Comment

Line 66: There is more literature on this topic, for example: Sun, Q., Miao, C., AghaKouchak, A., Mallakpour, I., Ji, D., & Duan, Q. (2020). Possible Increased Frequency of ENSO-Related Dry and Wet Conditions over Some Major Watersheds in a Warming Climate, Bulletin of the American Meteorological Society, 101(4), E409-E426. Retrieved Jun 17, 2021, from https://journals.ametsoc.org/view/journals/bams/101/4/bams-d-18-0258.1.xml

Reviewer 2 Report

The manuscript is interesting. It is well written. However, the manuscript is not yet ready for publication.

Authors should note in Introduction the differences between the dataset used. Rain gauge data refer to point measurements, while CHIRPS data (a gridded dataset) provide precipitation covering larger geographical areas.

CHIRPS is a high-resolution land-only climatic database of precipitation. CHIRPS involves three main components: i) the Climate Hazards group Precipitation climatology (CHPclim), ii) the satellite-only Climate Hazards group Infrared Precipitation (CHIRP), and iii) the station blending procedure that produces the CHIRPS. Thus, its should be pointed out in the manuscript that gauge data is also implemented in the CHIRPS database.

Authors should also comment how many of the gauge station data are used within CHIRPS database and if there are any station free CHIRPS pixels used in the analysis. Moreover, it would be more interesting if authors could make these comparisons by separating CHIRPS database to blended and non-blended gauge data.

Table 1: In legend it is referred: %N Rej and in the Table %Rej/T. Please check.

Round 2

Reviewer 1 Report

My suggestions and comments have been answered. The manuscript can now be published.

I suggest improving the quality of the supplementary figures, as was done with the figures in the manuscript.

Author Response

Thanks for your suggestions, we made the changes you suggested.

Oscar Mesa